# Transcriptome analysis revealed misregulated gene expression in blastoderms of interspecific chicken and Japanese quail F$_1$ hybrids

**Satoshi Ishishita[1], Shoji Tatsumoto[2], Keiji Kinoshita[1], Mitsuo Nunome[1], Takayuki Suzuki[1,3], Yasuhiro Go[2], Yoichi Matsuda** [1,3]*

1 Avian Bioscience Research Center, Graduate School of Bioagricultural Sciences, Nagoya University, Furo-cho, Chikusa-ku, Nagoya, Aichi, Japan, 2 Cognitive Genomics Research Group, Exploratory Research Center on Life and Living Systems (ExCELLs), National Institutes of Natural Sciences, Okazaki, Aichi, Japan, 3 Laboratory of Avian Bioscience, Department of Animal Sciences, Graduate School of Bioagricultural Sciences, Nagoya University, Furo-cho, Chikusa-ku, Nagoya, Aichi, Japan

* yoimatsu@agr.nagoya-u.ac.jp

**Data Availability Statement:** The Illumina data generated in the present study have been deposited in the DDBJ Sequence Read Archive

## Abstract

Hybrid incompatibility, such as sterility and inviability, prevents gene flow between closely-related populations as a reproductive isolation barrier. F$_1$ hybrids between chickens and Japanese quail (hereafter, referred to as quail), exhibit a high frequency of developmental arrest at the preprimitive streak stage. To investigate the molecular basis of the developmental arrest at the preprimitive streak stage in chicken–quail F$_1$ hybrid embryos, we investigated chromosomal abnormalities in the hybrid embryos using molecular cytogenetic analysis. In addition, we quantified gene expression in parental species and chicken- and quail-derived allele-specific expression in the hybrids at the early blastoderm and preprimitive streak stages by mRNA sequencing. Subsequently, we compared the directions of change in gene expression, including upregulation, downregulation, or no change, from the early blastoderm stage to the preprimitive streak stage between parental species and their hybrids. Chromosome analysis revealed that the cells of the hybrid embryos contained a fifty-fifty mixture of parental chromosomes, and numerical chromosomal abnormalities were hardly observed in the hybrid cells. Gene expression analysis revealed that a part of the genes that were upregulated from the early blastoderm stage to the preprimitive streak stage in both parental species exhibited no upregulation of both chicken- and quail-derived alleles in the hybrids. GO term enrichment analysis revealed that these misregulated genes are involved in various biological processes, including ribosome-mediated protein synthesis and cell proliferation. Furthermore, the misregulated genes included genes involved in early embryonic development, such as primitive streak formation and gastrulation. These results suggest that numerical chromosomal abnormalities due to a segregation failure does not cause the lethality of chicken–quail hybrid embryos, and that the downregulated expression of the genes that are involved in various biological processes, including translation and primitive streak formation, mainly causes the developmental arrest at the preprimitive streak stage in the hybrids.

(DRA003025). Gene expression data have been deposited in the DDBJ Gene Expression Achieve (E-GEAD-390). All further relevant data are within the manuscript and its Supporting Information files.

**Funding:** This work was supported by a Grant-aid for Scientific Research from MEXT, Japan (No. 23113004 for Innovative Areas "Correlative gene system" and No. 221S0002 for Innovative Areas "Genome Science") to YM. URL: https://www.mext.go.jp/en/index.htm The funders had no role in study design, data collection and analysis, decision to publish, or preparation of the manuscript.

**Competing interests:** No authors have competing interests.

## Introduction

Speciation, the process by which populations evolve into distinct species, is often associated with hybrid incompatibility, such as reduced fertility and viability of hybrid progeny [1–5]. Hybrid incompatibility may facilitate speciation by preventing gene flow between sympatric populations and also reinforce prezygotic reproductive isolation between populations through the promotion of the divergence of mating behavior or gametic interactions [6–8]. Hybrid incompatibility genes are defined as those that measurably decrease the fitness in F$_1$, F$_2$, or BC$_1$ generations of interspecific hybrids [9]. According to the Dobzhansky and Muller (DM) model [10–12], genetic diversification occurs at multiple loci in populations originating from the same ancestral population, and incompatible allelic interactions in hybrids cause hybrid abnormalities as by-products of the evolution: alleles that effectively function in pure-species genetic backgrounds may cause adverse effects in the genetic background of interspecific hybrids. Hybrid incompatibility genes have been identified in many interspecific or intersubspecific hybrids in a wide range of taxa, including *Saccharomyces*, *Arabidopsis*, *Oryza*, *Drosophila*, *Xiphophorus*, and *Mus* [9,13,14]. However, the molecular basis of hybrid incompatibility remains poorly understood.

Although chicken (*Gallus gallus domesticus*) and Japanese quail (*Coturnix japonica*) belong to different genera that diverged approximately 35 million years ago (MYA) [15], interspecific hybrids can be generated by artificial insemination (AI) of chicken semen into the quail oviduct [16,17]. However, most hybrids die before hatching, and only a few male hybrids, which account for approximately 4% of the fertilized eggs can hatch [18], which is consistent with Haldane's rule, "When in the F$_1$ offspring of two different animal races one sex is absent, rare, or sterile, that sex is the heterozygous (heterogametic) sex" [19–24]. Our previous observation of chicken–quail hybrid embryos revealed that hybrid lethality occurs at various stages of embryonic development, including blastoderm, somite, and postsomite stages [18]. Most hybrid embryos (approximately 75% fertilized eggs) displayed developmental arrest during extraembryonic membrane formation and blood island formation stages (0–2 days of incubation). Furthermore, a substantial fraction of the hybrid embryos incubated for 8.5–36 h died at the Eyal-Giladi and Kochav XI–XIV stages, which are known as the preprimitive streak stage [25]: hybrid embryos with abnormal morphology accounted for 48.1% at 21–36 h after starting incubation, and 46.2% of the abnormal embryos were arrested at the XI–XIV-like stages.

The primitive streak is an organizing center of gastrulation in amniotes [26]. During the preprimitive and subsequent primitive streak stages, cell migration, proliferation, and differentiation occur, resulting in the formation of the second body axis and three germ layers [27–29]. The abnormal morphologies of the stage XI–XIV-like blastoderms may be due to aberrant migration, proliferation, and/or differentiation of epiblast and/or hypoblast cells during the preprimitive and subsequent primitive streak stages. Early embryonic lethality has also been observed in other avian interspecific hybrids such as hybrids between chicken and ring-necked pheasant and between chicken and turkey [30–32]. Therefore, developmental arrest at the early embryonic stages may be a common feature of interspecific hybrids of Phasianidae. In mammals, males are the heterogametic sex; by contrast, the heterogametic sex is females in birds [23,33]. In addition, genomic imprinting has not been found in birds, unlike in mammals [34–36]. Thus, bird hybrids would provide new insight into the molecular basis of hybrid incompatibility.

Numerical chromosome abnormalities due to a failure of segregation of chromosomes is associated with embryonic lethality in some interspecific hybrids of fish, frogs, and other organisms [37–39]. Uniparental chromosome elimination has been demonstrated in hybrid cells of various organisms, including plants [40–42], insects [43,44], and mammals [45–49].

Several types of molecular processes, including transcriptional regulation, post-transcriptional regulation, and protein-protein interactions, may cause hybrid incompatibility phenotypes [50]. Inappropriate transcriptional regulation (overexpression and/or underexpression) is associated with lethality, abnormal growth, and sterility in hybrids of *Mus*, *Phodopus*, and *Xiphophorus* [51–54].

In the present study, to enhance our understanding of the cause of lethality at the preprimitive streak stage of chicken–quail hybrid embryos, we initially performed chromosome analysis of the hybrid embryos at the early blastoderm stage (stage X) and in 3-day and 7-day-old hybrid embryos. Subsequently, to investigate the molecular basis of the hybrid inviability, we performed gene expression analyses of the embryos at the stage X and preprimitive streak stages (stage XIII/XIV) for parental species and their F$_1$ hybrids. We generated expression profiles of genes at the two stages, determined the directions of expression changes (upregulation, downregulation, and no change) from the stage X to the stage XIII/XIV, and then compared the expression profiles between the hybrids and parental species.

## Materials & methods

### General

No statistical methods were used to predetermine sample size, and the experiments were not randomized. The investigators were not blinded to allocation during experiments and outcome assessment.

### Ethics statement

Animal care and all experimental procedures were approved by the Animal Experiment Committee, Graduate School of Bioagricultural Sciences, Nagoya University (approval number: 2018031334). Experiments were conducted in line with the Regulations on Animal Experiments at Nagoya University.

### Animals

Commercial quail were purchased from a local hatchery (Cyubu Kagaku Shizai, Nagoya, Japan), and fertilized chicken eggs of the Ehime-jidori (Japanese native chicken breed) [55] and the BL-E line (long-term closed colony of Brown Leghorn breed) [56] were supplied by the National BioResource Project Chicken/Quail, Nagoya University, Japan. For chromosome analysis, we used embryos of commercial quail and Ehime-jidori chickens and their F$_1$ hybrid embryos that were obtained by AI of semen from male Ehime-jidori chickens into female quail. Embryos of commercial quail, BL-E chickens and their F$_1$ hybrid embryos at stages X and XIII/XIV were used for gene expression analyses. The two analyses were conducted at different times using different chicken lines because of the availability of adult chickens of these lines when the analyses were carried out. Chickens and quail were maintained with free access to water and a commercially available diet. The photoperiod was 14:10 h L:D, and room temperature was maintained at approximately 25˚C. After all experiments, adult chickens and quail were decapacitated after isoflurane anesthesia.

### Artificial insemination (AI)

AI was performed twice a week. Chicken semen was collected just before AI from 5–10 adult males of each strain. After addition of gentamicin into pooled semen (final concentration of 10 μg/ml), 50–100 μl semen was injected into vaginas of quail using a syringe. To avoid the

excretion of the injected semen from vaginas by oviposition, AI was performed during the last 1–2 h of a light period, when oviposition on that day was completed in most female quail.

## Egg preservation and incubation

Laid eggs were stored at 12˚C until use. Eggs were used for incubation within 14 days of storage. They were incubated at 37.6˚C and 70% relative humidity, with rocking at an angle of 90˚ at 30-min intervals.

## mRNA sequencing

To extract total RNAs from blastoderms at the stage X, blastoderms were collected from the eggs that were laid on each day, which were preserved at 12˚C immediately after being laid. To extract total RNAs of blastoderms at the stage XIII/XIV, we began the incubation of the eggs within 3 d after they were collected and preserved at 12˚C, and blastoderms were collected after 7.5–10.0 h of incubation. We classified the developmental stages of hybrid blastoderms with abnormal morphology using the following criterion: hybrid blastoderms at stages similar to stages XIII–XIV of chickens, at which the hypoblast is formed, were considered stage XIII–XIV-like blastoderms.

After removal of egg yolk from blastoderms in phosphate buffered saline (PBS), the tissues were minced in a 5–10 μl of PBS by pipetting. One μl of each cell suspension was lysed in a 50 μl buffer containing 10 mM Tris-HCl pH 7.5, 5 mM EDTA, 0.5% Tween-20, and 50 μg/ml Proteinase K, and incubated at 50˚C for 15 min and then at 95˚C for 5 min. After centrifugation of the lysates at 12,000 rpm for 5 min, supernatants were used for molecular sexing, which was performed by PCR analysis of sequence length polymorphism of the intron of *CHD1* as described elsewhere [57]. Sequences of primers used for PCR were as follows: 2550F (5′-GTTACTGATTCGTCTACGAGA-3′) and 2718R (5′-ATTGAAATGATCCAGTGCTTG-3′). PCR products were visualized by 2% agarose gel electrophoresis. The 600-bp PCR fragment derived from the Z chromosome was detected in both sexes, whereas an additional W chromosomal 450-bp PCR fragment was amplified only in females.

The remaining cell suspensions were lysed in TORIZOL reagent (Life Technologies, Carlsbad, CA, USA) immediately after tissue sampling. The solutions including blastodermal tissues were transferred into QIAshredder Mini Spin Columns (Qiagen, Hilden, Germany) and spun down. The flow-through samples were stored at -80˚C until use. The frozen samples were thawed on ice, and total RNAs were purified according to the manufacturer's instructions. The aqueous phases were transferred into Buffer RLT of RNeasy Plus Micro Kit (Qiagen) and then total RNAs were purified. RNA quality was assessed using Bioanalyzer Pico Chips (Agilent Technologies, Santa Clara, USA). RNAs whose RNA Integrity Numbers were over 7.5 were used for mRNA sequencing.

We converted oligo(dT)-selected RNA into cDNA libraries for mRNA sequencing using the SureSelect Strand Specific RNA Library preparation kit (Agilent Technologies) according to the manufacturer's instructions. The libraries were sequenced on an Illumina® HiSeq 2500 platform using paired-end sequencing (100 bp). A total of 174 GB was obtained from 48 libraries (average of 3.6 GB per sample). We trimmed the adapter sequences from the reads using Trimmomatic v0.33 [58], and then mapped the reads to the reference genome (Accession codes: GCF_000002315.5 for chickens and GCF_001577835.1 for quail) using HISAT2 v2.1.0 [59]. Multi-mapped reads and reads with >2 mismatches were filtered out using SAMtools v1.9 [60], and orphan reads were eliminated using a custom Perl script. Read counts per gene were calculated by HTSeq v0.11.2 [61] using concordantly aligned read pairs. For the analysis of allelic expression in the hybrids, reads that were mapped to both reference genomes of

parental species were removed before the calculation of read counts per gene using Bash scripts. Before detecting differentially expressed genes between stage X and stage XIV embryos, we excluded the genes whose counts fell below the threshold (1) in any sample in the dataset. Afterward, we used the Wald test for significance testing using DESeq2 v1.18.1 [62]. Fold change (FC) of gene expression was calculated by comparing gene expression between stage X and stage XIV embryos using DESeq2 v1.18.1. *P* values were adjusted using the Benjamini–Hochberg method. Genes with adjusted *P* value (false discovery rate, FDR) less than 0.05 and FC more than 2 were considered upregulated and those with FDR less than 0.05 and FC less than 0.5 were considered downregulated. Genes with FDR more than or equals to 0.05 or FC more than or equals to 0.5 and less than or equals to 2 were considered unaltered.

## GO term enrichment analyses

Gene Ontology (GO) term enrichment analyses were performed using the overrepresentation test (Released 20200728) of the PANTHER (Protein ANalysis THrough Evolutionary Relationships) Classification System [63]. The *Gallus gallus* database was used as the reference (GO Ontology database doi: 10.5281/zenodo.3954044). *P* values of Fisher's exact test were adjusted using the Benjamini–Hochberg method. GO terms were considered significant if they had an FDR less than 0.05. We referred to the AmiGO 2 database (v2.5.13) for the relationship of GO terms [64,65].

## Cell culture and chromosome analysis

We prepared chromosomes from stage X blastoderms according to methods described previously [66]. Briefly, blastoderms were incubated in Hank's solution at 39˚C for 1 h, incubated in hypotonic solution (0.9% sodium citrate) at room temperature (RT) for 30 min, and then fixed in 3:1 methanol: acetic acid fixative at RT for 30 min. After the removal of the fixative, tissues were suspended in 50% acetic acid at RT for 5–10 min. After pipetting gently, 5–10 μl of cell suspension was spread on glass slides on a hot plate at 50˚C. The preparations were stained using 4% Giemsa solution for 10 min.

Chromosomes from metaphase nuclei were also prepared from cultured fibroblast cells derived from 3-day-old female embryos of chickens and quail, and 3-day and 7-day-old male and female hybrid embryos. The embryonic fibroblast cells were cultured in Dulbecco's modified Eagle's medium (Invitrogen-GIBCO, Carlsbad, CA, USA) supplemented with 15% fetal bovine serum (Invitrogen-GIBCO), 100 μg/ml kanamycin, and 1% Gibco® Antibiotic–Antimycotic (PSA) (Invitrogen-GIBCO) at 39˚C under 5% $CO_2$. Replication-banded chromosome slides were prepared for *in situ* hybridization as described previously [67]. The fibroblast cell cultures were treated with 5-bromodeoxyuridine (BrdU) (25 μg/ml) (Sigma-Aldrich, St Louis, MO, USA) at the late replication stage for 4.5 h including 0.5-h colcemid treatment. After staining the slides with Hoechst 33258 (3 μg/ml) for 5 min, they were heated at 65˚C for 3 min and exposed to UV light at 65˚C for 6.5 min. The slides were stored at -80˚C until use.

## Fluorescence *in situ* hybridization

Fluorescence *in situ* hybridization of centromeric DNA repeats and chromosome painting with DNA probes of chicken chromosomes 1–8 and Z [68] were conducted as described previously [69]. The DNA probes were labeled with biotin-16-dUTP (Roche Diagnostics, Basel, Switzerland) by nick translation and hybridized to metaphase spreads at 37˚C for 4 days. After washing, the slides were incubated with FITC-avidin (Roche Diagnostics). For dual-color FISH, the biotin- and DIG-labeled probes were reacted with FITC-avidin and anti-DIG antibody (Roche Diagnostics), respectively [70].

## Imaging

We used a cooled charge-coupled device (CCD) camera (Leica DFC360 FX, Leica Microsystems, Wetzlar, Germany) mounted on a Leica DMRA microscope for FISH and chromosome painting, and analyzed the data using the 550CW-QFISH program (Leica Microsystems Imaging Solutions Ltd., Cambridge, UK).

## Statistical analysis

R v.3.4.3 (R Core Team) and MS Excel (Microsoft Corp., Redmond, WA, USA) were used for statistical analyses. In addition, we used the Tukey-Kramer test for pairwise comparisons of total number of chromosomes and for the comparison of the number of microchromosomes between parental species and chicken–quail hybrids. A $P$-value less than 0.05 indicated statistical significance. We also calculated Pearson's correlation coefficient for correlational analyses of changes in gene expression.

## Results

### Chromosome analysis of embryos of chicken, quails, and their F₁ hybrids

The number of chromosomes in both quail and chicken is 78 (2 n = 78) [71]. We prepared chromosomes from stage X blastoderms and 3-day and 7-day-old embryos of chicken, quails, and their F₁ hybrids (Fig 1A) for molecular cytogenetic analyses of hybrid nuclei. Macrochromosomes consisted of nine pairs of homologous chromosomes including ZZ or ZW sex chromosomes, some of which exhibited slight differences in size between chicken and quail chromosomes (Fig 1B). Chromosome painting with chicken macrochromosome-specific (chromosomes 1 to 8) and Z chromosome DNA probes [68], and hybridization with a chicken W-specific DNA repeat [72] confirmed that hybrid nuclei consisted of nine pairs of macrochromosomes and one pair of sex chromosomes (Fig 1C). One of each macrochromosome pair exhibited a larger size and stronger hybridization signal than the other, which suggested that the larger-sized chromosome in each pair originated from chicken. All the observed nuclei (96–158 nuclei for one DNA probe) showed two painted signals for each chromosome-specific DNA probe or one hybridization signal for W-specific DNA repeat (S1 Table).

Subsequently, we attempted to discriminate the parental origins of microchromosomes on metaphase spreads of the hybrid nuclei. Centromeric DNA repeats that are predominantly localized to microchromosomes have been isolated from chicken (GGA-TaqI-8) and quail (CJA-BglII-M9) in previous studies [73,74]. GGA-TaqI-8 and CJA-BglII-M9 were hybridized into most of the microchromosomes derived from chicken and quail, respectively (Fig 2A and 2B), although GGA-TaqI-8 was also hybridized into two pairs of macrochromosomes (Fig 2A). We observed intense hybridization signals of GGA-TaqI-8 on chicken microchromosomes (Fig 2A) and very weak cross-hybridization signals of this probe on a part of quail microchromosomes (Fig 2B). CJA-BglII-M9 exhibited non-species-specific hybridization; the repeat was hybridized into quail microchromosomes (Fig 2B) and cross-hybridized into a part of chicken microchromosomes and a few macrochromosomes (Fig 2A). Therefore, CJA-BglII-M9 detected quail microchromosomes and chicken microchromosomes simultaneously (Fig 2C). We considered microchromosomes that were hybridized with GGA-TaqI-8 or with both of the two repeats as chicken-derived chromosomes, and those exhibiting hybridization signals of CJA-BglII-M9, with no or weak cross-hybridization signals of GGA-TaqI-8, as quail-derived chromosomes.

We counted the number of chromosomes on metaphase spreads of the 0-h blastoderms and 3- and 7-day-old embryos. The total number of chromosomes in the hybrid cells was

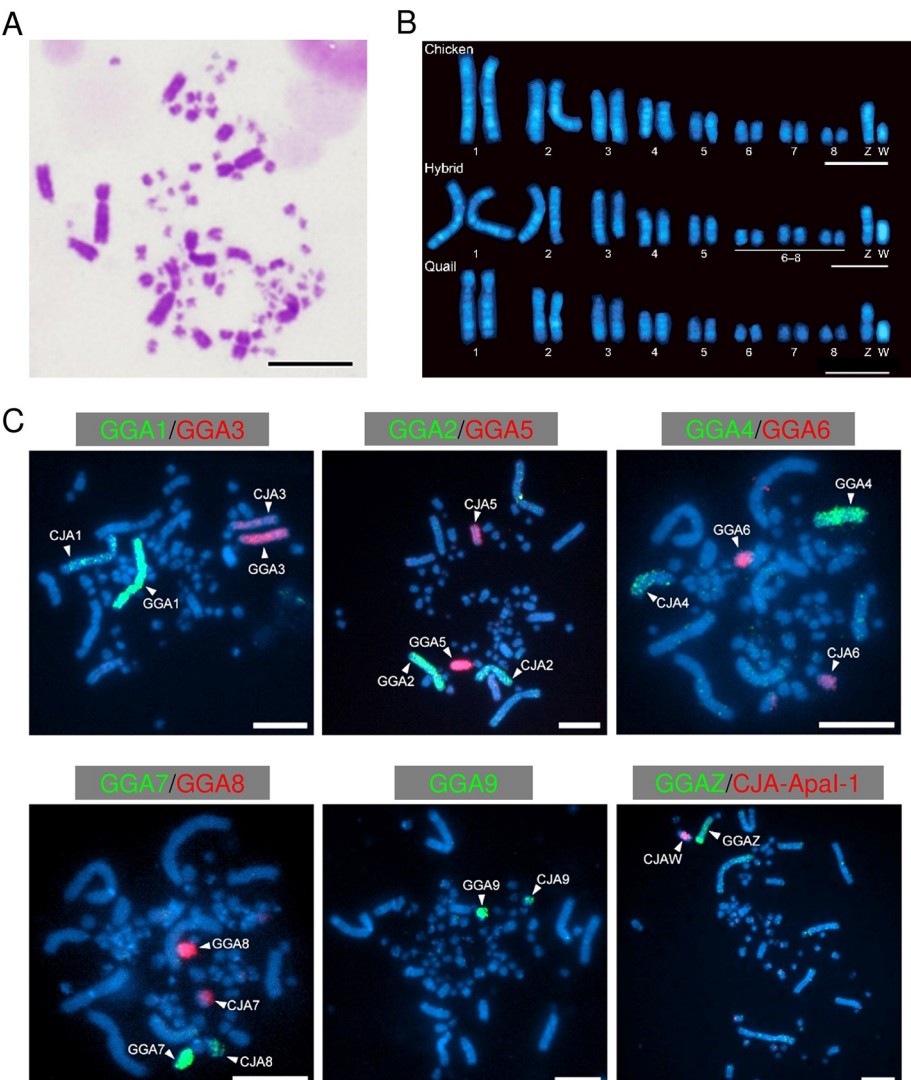

**Fig 1. Chromosome analysis of chicken–quail F$_1$ hybrids.** A. Giemsa-stained metaphase spread of a blastodermal cell of the chicken-quail F$_1$ hybrid, consisting of large-sized macrochromosomes and small-sized microchromosomes. B. Hoechst-stained chromosomes of cultured fibroblast cells from embryos of the chicken, chicken–quail hybrid, and quail, which show eight pairs of macrochromosomes and the Z and W sex chromosomes. The sizes of each pair of chromosomes differ between the chicken and quail chromosomes in the hybrid. C. Chromosome painting with macrochromosome-specific DNA probes and hybridization with the W-specific DNA repeat in fibroblast cells of the hybrids. Larger macrochromosomes with stronger hybridization signals are considered to be derived from chicken (GGA, *Gallus gallus*), and the others from quail (CJA, *Coturnix japonica*). Scale bars, 10 μm.

mostly 78, which was nearly equal to that in parental species (Fig 3A and 3B, S1 Data, Tukey-Kramer test, $P > 0.05$). We counted chicken- and quail-derived microchromosomes on metaphase spreads of the hybrids using two repeated sequences. The numbers of chicken- and quail-derived microchromosomes with positive signals in the hybrids were 25 and 23 on average, respectively, which were nearly equal to half the number of microchromosomes that could be detected with the chicken- and quail-specific centromeric repeats (Fig 3C and 3D, S1 Data, Tukey-Kramer test, $P > 0.05$). The total number of microchromosomes was 58 (29 pairs) in both parental species; therefore, our method using centromeric DNA repeats could not fully discriminate the parental origins of microchromosomes. However, the results

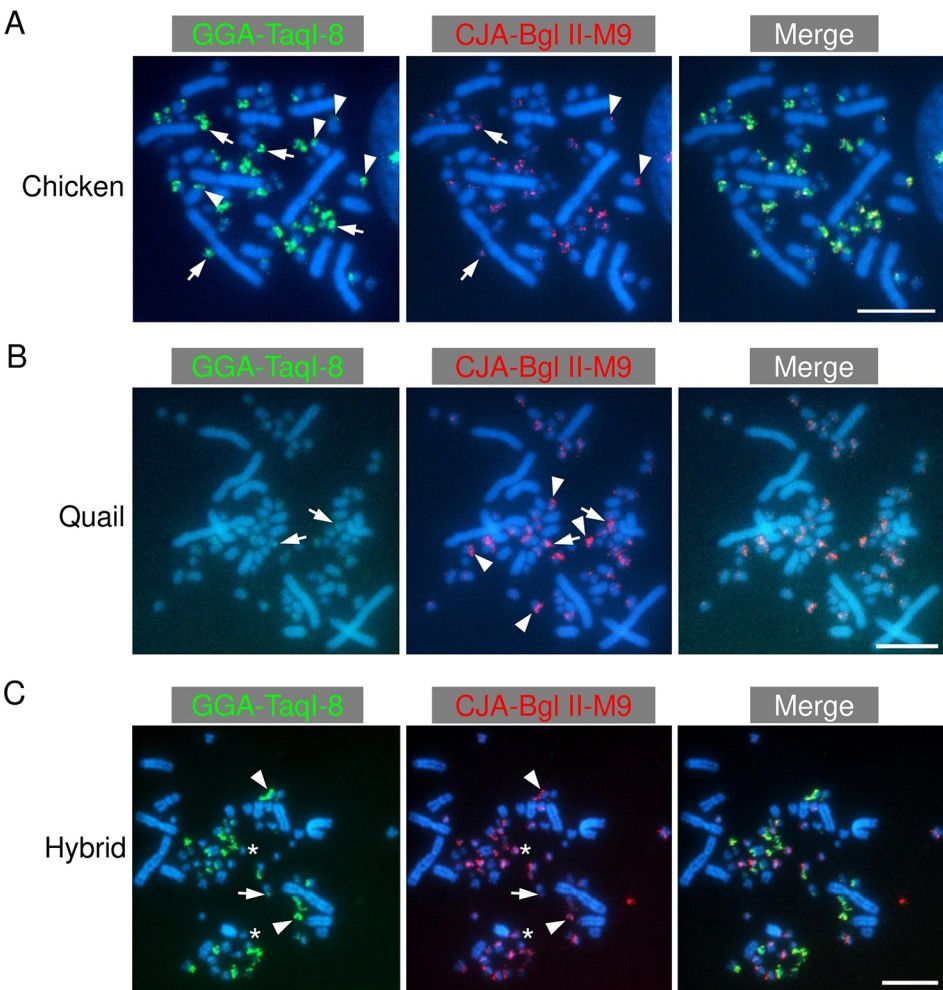

**Fig 2. Chromosomal localization of chicken and quail centromeric repetitive sequences in chickens, quail, and their hybrids.** Fluorescence-labelled DNA probes of chicken and quail centromeric repeats (GGA-TaqI-8 and CJA-BglII-M9, respectively) were hybridized into chromosome spreads of chickens (A), quail (B), and their hybrids (C). A. GGA-TaqI-8 was localized to almost all microchromosomes (arrows in the left panel indicate representatives) and two pairs of macrochromosomes (arrowheads in the left panel). CJA-BglII-M9 was cross-hybridized into a part of microchromosomes and a few macrochromosomes (arrows and arrowheads, respectively, in the middle panel). B. GGA-TaqI-8 was cross-hybridized into quail chromosomes, which was observed as weak hybridization signals (arrows in the left panels). CJA-BglII-M9 was localized to almost all microchromosomes (arrows and arrowheads in the middle panel). C. Chromosomes that were hybridized only with GGA-TaqI-8 (arrows) and those that were hybridized with both GGA-TaqI-8 and CJA-BglII-M9 (arrowheads) were observed in the hybrid. These were considered as chicken-derived chromosomes. In addition, chromosomes that exhibited hybridization signals of CJA-BglII-M9, with no or weak hybridization signals of GGA-TaqI-8, were observed (asterisks). These were considered as quail-derived chromosomes. Scale bars, 10 μm.

collectively suggest that the hybrid nuclei consist of a fifty-fifty mixture of chicken and quail chromosomes, and that numerical abnormality, such as chromosome loss and/or duplication, hardly occurred in the hybrids.

## Expression changes in chicken and quail genes in the hybrids and the parental species

To study the molecular basis of developmental arrest in chicken–quail hybrid embryos at the preprimitive streak stage (Fig 4A and 4B), we performed whole-transcriptome analyses of the

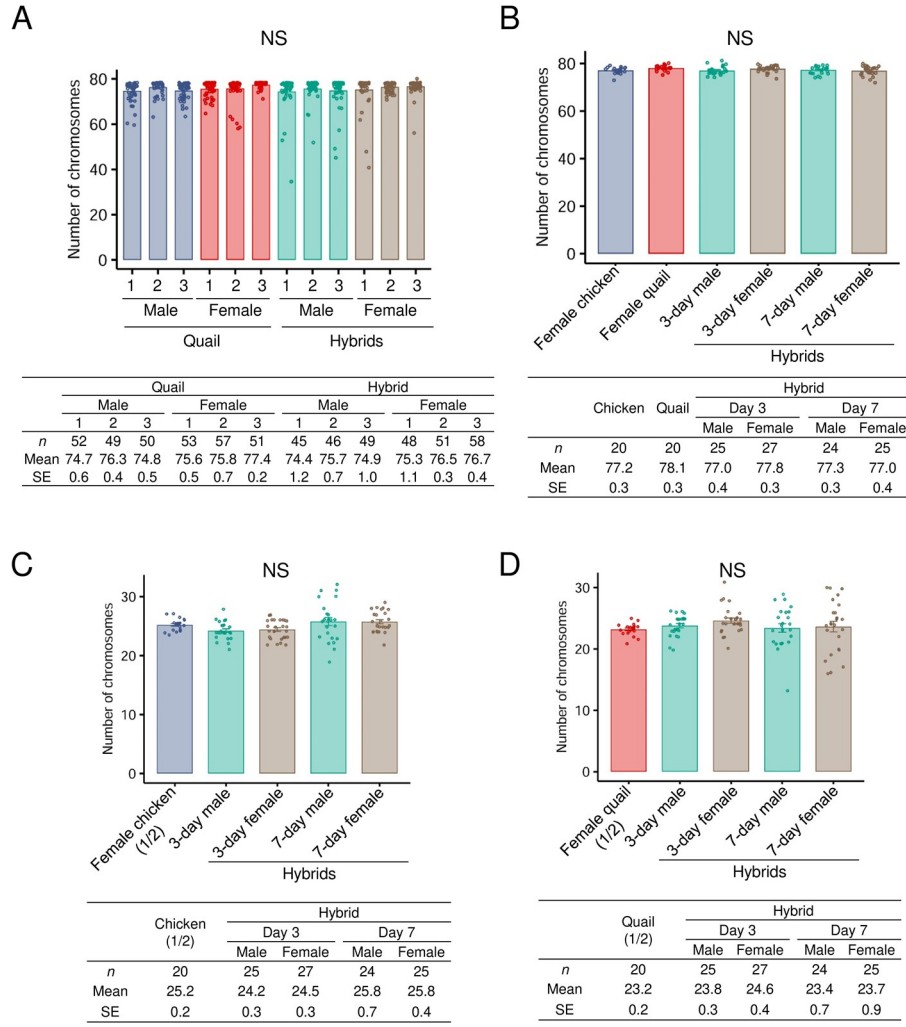

**Fig 3. Number of chromosomes in parental species and their F$_1$ hybrids.** A. Total number of chromosomes in blastodermal cells of quail and the hybrids. Three males and three females were used for each group. The number of chromosomes was not different among individuals (Tukey-Kramer test, $P > 0.05$). NS, not significant. B. Total number of chromosomes in fibroblast cells from 3- or 7-day-old male and female hybrid embryos and those from 3-day-old female embryos of parental species. The number of chromosomes was not different among them (Tukey-Kramer test, $P > 0.05$). C. The number of chromosomes hybridized with GGA-TaqI-8 in 3-day and 7-day-old male and female hybrid embryos. In 3-day female chickens, a half of the total number of microchromosomes per nucleus, which were detected with GGA-TaqI-8, is shown. The number of chicken-derived microchromosomes in the hybrids did not deviate from half the number of GGA-TaqI-8-positive microchromosomes in chicken cells (Tukey-Kramer test, $P > 0.05$). D. The number of chromosomes hybridized with CJA-BglIII-M9 in 3-day and 7-day-old male and female hybrid embryos. In 3-day female quail, a half of the total number of microchromosomes per nucleus, which were detected with CJA-BglIII-M9, is shown. The number of quail-derived microchromosomes in hybrid cells did not deviate from half the number of CJA-BglII-M9-positive microchromosomes in quail cells (Tukey-Kramer test, $P > 0.05$).

embryos of parental species and their F$_1$ hybrids at stages X and XIII/XIV by mRNA sequencing (Fig 4C, S1 and S2 Figs). Average numbers of mapped reads in the stage X embryos were 31.1 million in chickens, 26.5 million in quails, and 8.9 million for chicken-derived alleles and 11.1 million for quail-derived alleles in the hybrids, and 17.7 million in chickens, 18.3 million in quail, and 7.5 million for chicken-derived alleles and 8.9 million for quail-derived alleles in the stage XIII/XIV embryos. We used species-specific reads to avoid apparent increases in

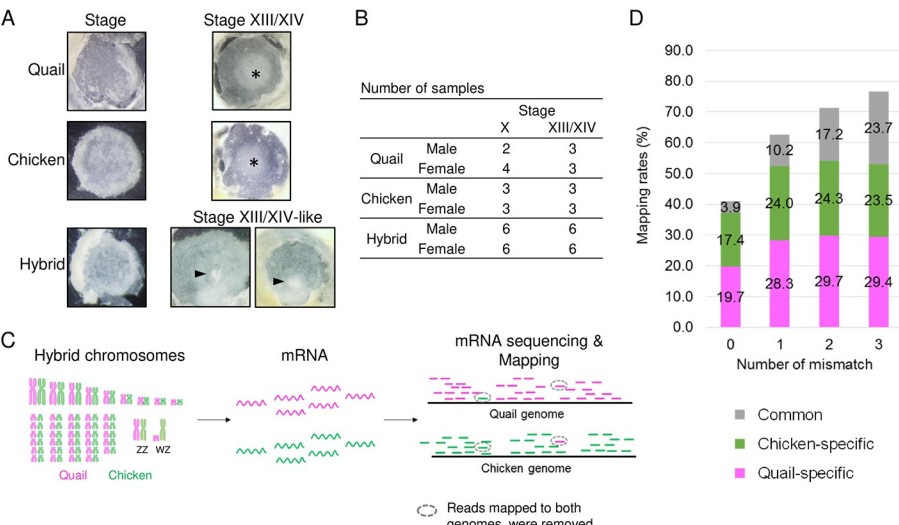

**Fig 4. Experimental scheme of gene expression analysis.** A. Quail, chicken, and hybrid embryos at stages X and XIII/XIV. Stage XIII embryos of parental species showed hypoblast cells in the middle region (asterisks). Hybrid embryos showed extensively proliferated hypoblast cells (arrows). B. Numbers and sexes of embryo samples used for mRNA sequencing. C. Schematic diagrams of mRNA sequencing and read mapping. Sequence reads from the hybrids were mapped to reference genomes of parental species. The reads that were mapped to both reference genomes were removed before counting the number of mapped reads for each gene. D. Percentage of reads that were mapped to chicken reference genome (chicken-specific), quail reference genome (quail-specific), or both genomic sequences (common). Number of mismatches indicate the maximum number of allowed mismatches per read.

gene expression owing to redundant mapping of the reads that were mapped to both reference genomes. Fig 4D shows mapping rates of sequence reads in a male hybrid at the XIII/XIV-like stage. The number of species-specific reads was the maximum when two mismatches were allowed in read mapping. Therefore, we estimated allelic expression of genes using species-specific reads obtained under such mapping conditions.

Relative gene expression levels cannot be compared directly between chicken and quail because the efficiency of read mapping is considered to vary between two species owing to differences in the reference genome sequences. Therefore, we investigated changes in gene expression from the stage X to the stage XIII/XIV for chicken (G) and quail (Q) genes in each parental species and chicken-derived alleles (HG) and quail-derived alleles (HQ) in the hybrids, and then compared changes in expression between G and Q, between G and HG, and between Q and HQ (S2 Data). The correlation coefficients of the expression change were much higher in Q vs. HQ (0.521) and G vs. HG (0.537) than in G vs. Q (0.107) in males (Fig 5A–5C). Similar results were also obtained from transcriptome analysis of female embryos (S3 Fig). We then determined the directions of change in expression (upregulated, downregulated, or unaltered expression) for each gene by differential gene expression analysis between the stage X and XIII/XIV male blastoderms (Fig 5D). We revealed that 8,376 (72.4%) out of a total of 11,575 genes exhibited similar changes in expression between quail (Q) and chickens (G) (Fig 5E). We also showed that 9,253 (79.9%) and 8,572 (74.1%) genes exhibited similar changes in expression between quail (Q) and hybrids (quail-derived alleles, HQ) and between chickens (G) and hybrids (chicken-derived alleles, HG), respectively, in males (Fig 5F and 5G). The numbers were higher than that between parental species. Similar results were also obtained in females (S3 Fig). The results suggest that expression profiles of chicken and quail alleles in the hybrids retain considerable patterns of gene expression from the parental species.

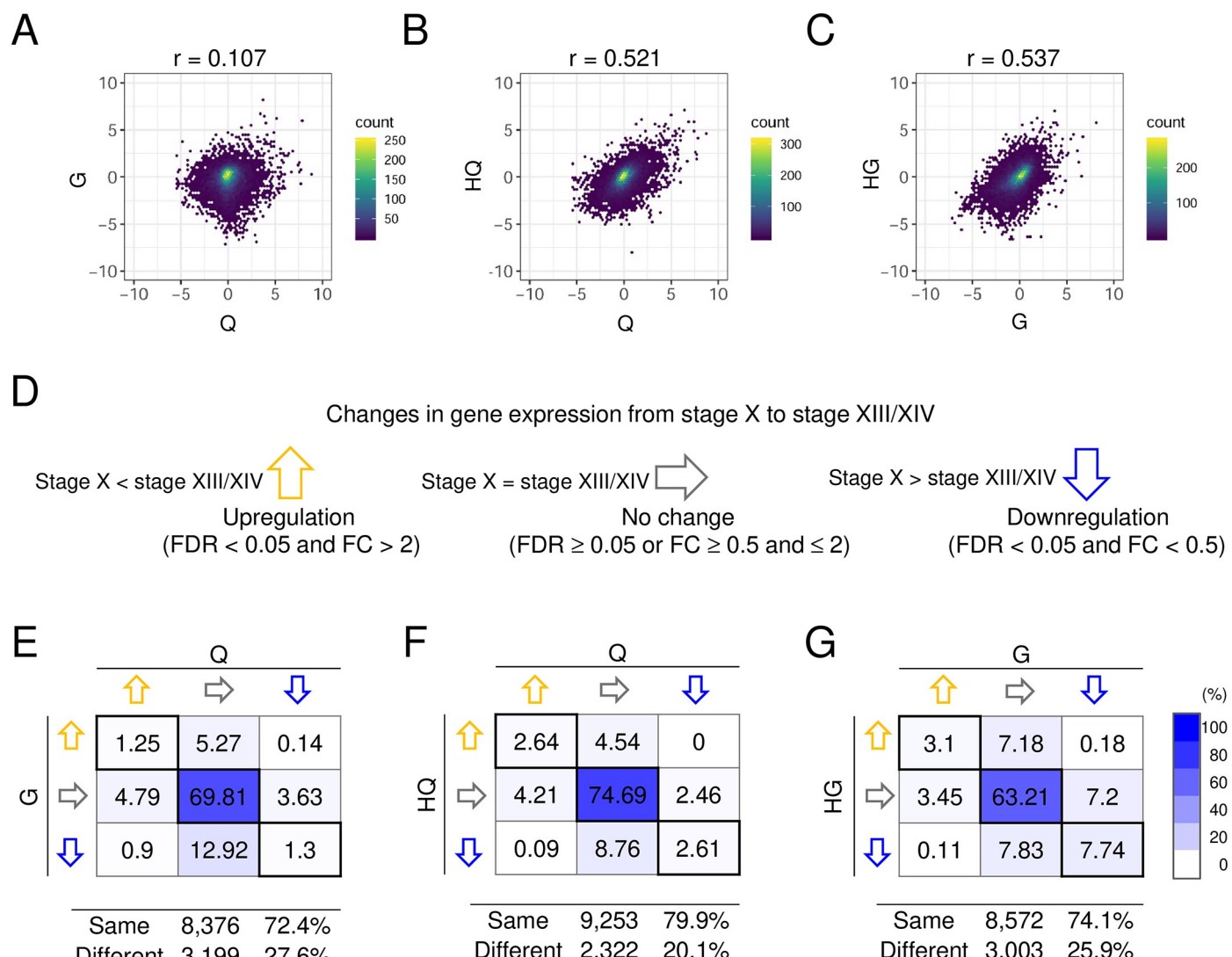

**Fig 5. Gene expression changes from the stage X to the stage XIII/XIV in male embryos.** A–C. Comparison of gene expression changes [log2(fold change)] between quail ('Q') and chickens ('G') (A), between quail ('Q') and quail-derived alleles in the hybrids ('HQ') (B), and between chickens ("G") and chicken-derived alleles in the hybrids ('HG') (C). Pearson's correlation efficient (r) is indicated above the graphs. D. Pattern classification of gene expression changes from the stage X to the stage XIII/XIV. E–G. Comparison of the direction of gene expression changes between quail and chickens (E), between quail and quail-derived alleles in the hybrids (F), and between chickens and chicken-derived alleles in the hybrids (G). The number in each rectangle indicates the percentage of genes. Percentages of genes that exhibited the same directions of expression changes are indicated in bold-lined rectangles. Numbers and percentages of genes exhibiting the same or different directions of expression change are shown in the tables. Color scale at the far right shows the percentage of genes.

## Identification of candidate genes responsible for developmental arrest

We postulated that genes whose expression is upregulated from the stage X to the stage XIII/XIV in both parental species play essential roles in the developmental process of embryos; the downregulated or unaltered expression (hereafter, referred to as misregulated expression) of such genes could cause the developmental arrest in the hybrid embryos. Subsequently, we searched for genes exhibiting misregulated expression in the hybrids (referred to as pattern D in Fig 6A). We found that 285 genes were upregulated from the stage X to the stage XIII/XIV in male and/or female embryos of parental species (S3 Data). Out of these 285 genes, 60 exhibited misregulated expression in males and/or females (S2 Table; Fig 6A, S4 Fig). Only four

genes, encoding BMP binding endothelial regulator (BMPER), gap junction protein alpha 1 (GJA1, also known as Connexin43), ribosomal protein SA (RPSA), and Wnt family member 5B (WNT5B), exhibited pattern D of expression in both sexes.

GO term enrichment analysis using PANTHER [63] revealed that 23 GO terms of biological process (hereafter, referred to as GO-BP terms) were overrepresented (FDR < 0.05) in the 285 genes whose expression was upregulated in both parental species (S3 Table). We show GO-BP terms that are related to primitive streak formation and chromosome segregation in S4 Table. Although none of these GO-BP terms were significantly overrepresented, the 285 genes included genes that are associated with primitive streak formation (S5 Table). GO term enrichment analysis of the 60 misregulated genes showed 14 overrepresented GO-BP terms (FDR < 0.05); 11 of these GO-BP terms, including peptide biosynthetic process, translation, amide biosynthetic process, nitrogen compound biosynthetic process, and ribosome biogenesis, involved numerous ribosomal protein genes (indicated in gray in Fig 6B and S6 Table). The other overrepresented GO-BP terms included regulation of sprouting angiogenesis and cell population proliferation. No GO-BP terms related to the formation of primitive streak (S4 Table) were not nominated in this enrichment analysis; however, several of the 60 misregulated genes were associated with primitive streak formation-related GO-BP terms, including gastrulation and anterior/posterior axis specification (Fig 6C; S5 Table). One of the 60 genes, *HORMAD2*, was associated with chromosome segregation-related GO-BP terms (S6 Table); however, *HORMAD2* has been known to play a role in meiosis, not mitosis [75]. Thus, the finding is consistent with the finding from chromosome analyses in the present study, in that chromosome segregation may not be affected hybrid embryos.

We then examined the expression of nine genes that are widely used as molecular markers of embryonic polarity before, during, and after primitive streak formation (Fig 7A) [27,76–85]. Most of the genes displayed similar changes in expression between parental species and their F₁ hybrid males (Fig 7B) and females (S5 Fig). Only *WNT5B* exhibited the pattern D of gene expression in both sexes (Fig 7B, S5 Fig).

## Discussion

Elucidation of the molecular basis of hybrid incompatibility in birds may enhance our understanding of their speciation process. In the present study, we investigated the cause of lethality of chicken–quail F₁ hybrid embryos at the preprimitive streak stage by focusing on chromosome segregation and gene expression. This is, to the best of our knowledge, the first study investigating the cause of lethality of chicken-quail F₁ hybrids by molecular cytogenetic analysis and mRNA sequencing. We have demonstrated that nuclei of the chicken–quail hybrid embryos had a fifty-fifty mixture of parental chromosomes and that numerical abnormalities due to a failure in chromosomal segregation hardly occurred, which is consistent with the observations of previous cytological studies [86,87]. In the present study, we examined only 0-h-old blastoderms and cultured fibroblast cells from 3-day and 7-day-old embryos; therefore, it remains unclear whether chromosome abnormalities occur at different developmental stages and/or in different types of cells in the hybrids. However, the results of the present study suggest that numerical chromosomal abnormalities due to a segregation failure, which have been observed for interspecific hybrids of other organisms [37–39], were not a major cause of the lethality in the chicken–quail hybrid embryos.

Transcriptome analysis of stage X and stage XIII/IV embryos revealed that out of the genes whose expression was upregulated from the stage X to the stage XIII/XIV in the embryos of parental species, neither chicken- nor quail-derived alleles were upregulated for 60 genes in the hybrid males and/or females. Such misregulated genes are potentially responsible for

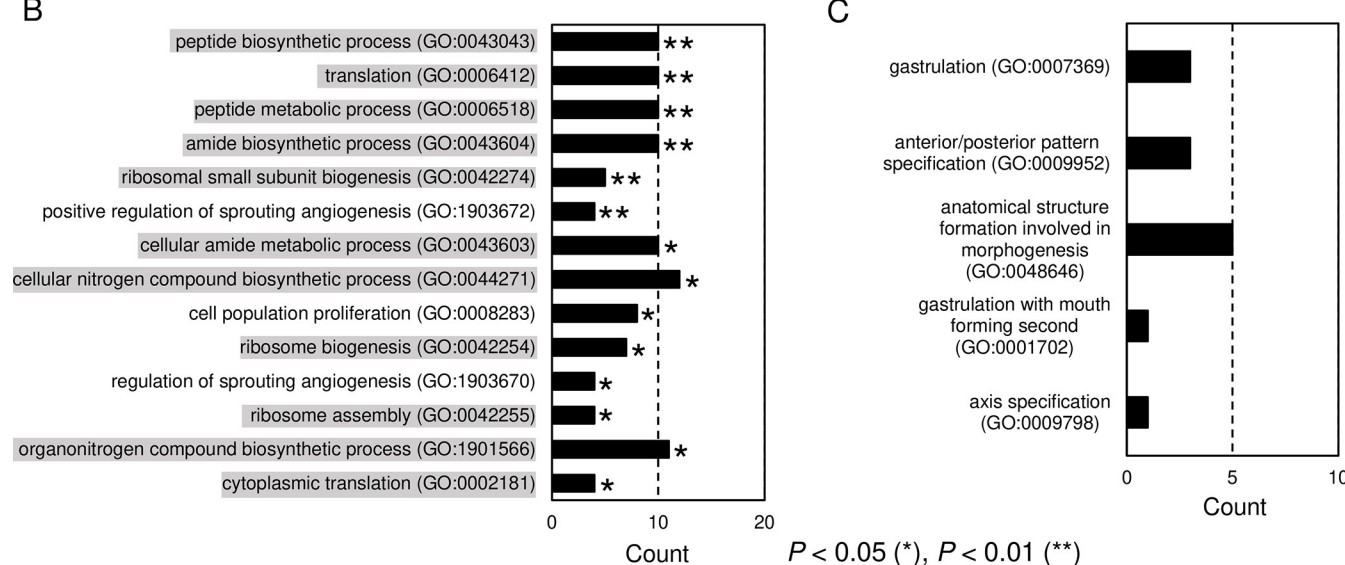

**Fig 6. GO term enrichment analysis of genes whose expression was misregulated in the male and/or female hybrids.** A. Number of genes whose expression was upregulated from the stage X to the stage XIII/XIV in male and/or female embryos of parental species. In pattern D, 60 genes showed no upregulation of their chicken- and quail-derived alleles in male and/or female hybrids. B. Overrepresented GO-BP terms and the number of genes that are associated with these terms. Eleven GO-BP terms that involve numerous ribosomal protein genes are indicated in gray. C. GO-BP terms related to primitive streak formation and the number of genes associated with these terms.

developmental arrest at the preprimitive streak stage. GO term enrichment analysis of the 60 misregulated genes revealed that GO-BP terms, including peptide biosynthetic process, translation, cell population proliferation, and sprouting angiogenesis, were significantly overrepresented. These results suggest that biological processes, such as translation and expansion of cell population, could be affected considerably in the hybrids, with adverse effects on cell migration, proliferation, and/or differentiation in the embryos at the preprimitive streak stage, resulting in developmental arrest. Many ribosomal protein genes were misregulated in the hybrids, suggesting the presence of incompatibilities between chicken- and quail-derived genetic elements that regulate the expression of these ribosomal protein genes. It is unconceivable that the developmental arrest is caused by abnormal sprouting angiogenesis because blood vessels are not formed at the preprimitive streak stages [88].

**A**

| Gene | Function | References |
|---|---|---|
| *WNT8A* | induces *NODAL* expression with other signaling factors | Bertocchini et al. 2004 [27] |
| *PITX2* | functions upstream of signaling factors that induces *NODAL* | Torlopp et al. 2014 [76] |
| *NODAL* | regulates the induction and/or maintenance of the primitive streak | Conlon et al. 1994 [77] |
| *CHRD* | promotes primitive streak formation through inhibiting BMP signaling | Streit et al. 1998 [78] |
| *WNT5A/5B* | regulates cell migration through the primitive streak | Hardy et al. 2008 [79] |
| *TBXT* | functions in the genesis of mesoderm during gastrulation and in the maintenance of a notochord | Kispert et al. 1995 [80]; Knezevic et al. 1997 [81] |
| *GSC* | is involved in the development of the organizer region Hensen's node | Izpisúa-Belmonte et al. 1993 [82] |
| *CNOT1* | is involved in Hensen's node, notochord and neural plate formation | Stein and Kessel 1995 [83] |

**B**

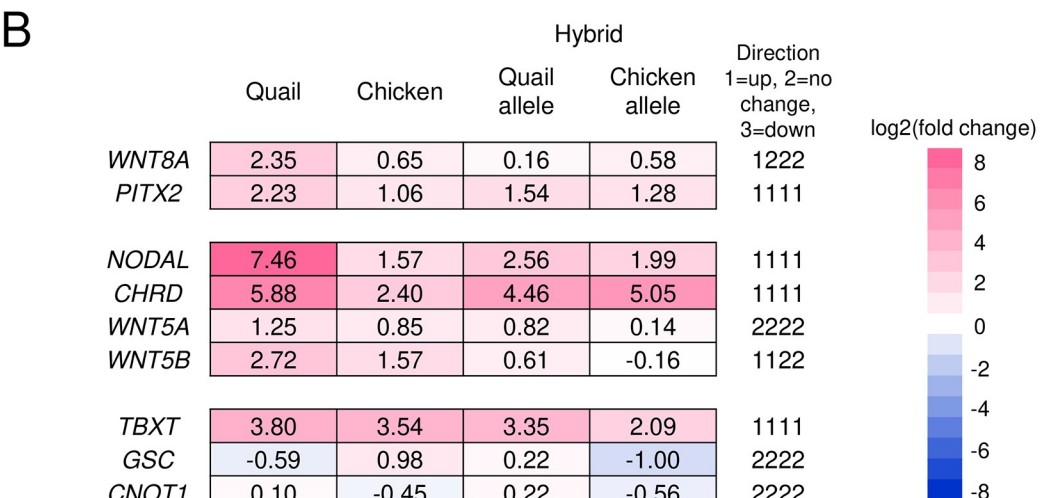

Fig 7. **Expression changes of primitive streak formation-related genes.** Genes that function before (*WNT8A* and *PITX2*), during (*NODAL*, *CHRD*, and *WNT5A/5B*), and after (*TBXT*, *GSC*, and *CNOT1*) the formation of the primitive streak (A) and changes of their expression from the stage X to the stage XIII/XIV in male embryos (B). Four-digit numbers shown on the right side of the column indicate the direction of expression changes in quail, chickens, and their hybrids (quail- and chicken-derived alleles), respectively. Only *WNT5B* displays the pattern D of gene expression. Color scale on the far right shows the degree of gene expression change.

Alternatively, the developmental arrest may result from the dysregulation of other biological processes. Out of the 60 genes, there were several genes that are associated with primitive streak formation-related GO-BP terms, such as gastrulation and anterior/posterior axis specification. For instance, *CRB2* is essential for normal mesoderm formation and is involved in the ingression of epiblast cells during the epithelial-to-mesenchymal transition at gastrulation [89]. *WNT3A* may mediate the formation of paraxial mesoderm in the anterior primitive streak [90]. *RPS6* haploinsufficiency induces embryonic death during gastrulation in mice [91]. Furthermore, out of the well-known primitive streak formation-related genes, *WNT5B* expression was misregulated in hybrids of both sexes. *WNT5B* is required for normal cell migration through primitive streak during gastrulation [79]. Expression of *WNT8A* and *PITX2*, which are involved in the initiation of primitive streak formation [27, 76], was not misregulated in the hybrids. Therefore, the misregulated expression of the genes involved in

primitive streak formation and gastrulation could inhibit the formation, but not the initiation, of the primitive streak, which may block the normal progression of primitive streak formation, resulting in the developmental arrest at the preprimitive streak-like stage.

In addition to *WNT5B*, *BMPER*, *GJA1*, and *RPSA* exhibited pattern D expression in both sexes. BMPER functions in gastrulation though inhibiting BMP signaling [92]. GJA1 mediates gap junctional communication for morphogenesis during gastrulation [93]. Therefore, the misregulated expression of *BMPER* and *GJA1* may also cause the developmental arrest at the preprimitive streak-like stage through inhibiting gastrulation. RPSA is a component of the 40S subunit and also acts as a membrane receptor [94]. The protein is required for pre-rRNA processing and spleen formation in *Xenopus* [95]; however, its role in gastrulation remains unknown.

Male sterility in *M. m. musculus* × *M. m. domesticus* hybrids is one of the most intensively studied models of hybrid incompatibility [96,97]. The hybrid male sterility in *Mus musculus* subspecies arises from abnormal expression of X-linked genes in testes, which is caused by incompatibility between X chromosome-linked genes and autosomal genes [53,98–101]. Therefore, misregulated gene expression in chicken-quail F$_1$ hybrids may also be caused by incompatible interaction between chicken- and quail-derived genes in hybrid embryos. Further investigation of the causal relationship of the 60 misregulated genes to developmental arrest in the hybrid embryos and the molecular mechanisms of misregulated expression of these genes may provide important information to understand the molecular basis of hybrid incompatibility in birds.

## Conclusions

We hypothesized that chromosomal abnormality is associated with the developmental arrest of chicken–quail F$_1$ hybrid embryos; however, it may be not the case considering the findings of the present study. We revealed misregulated expression of genes that are involved in various biological processes including translation, cell proliferation, and gastrulation as a potential cause of developmental arrest at the preprimitive streak-like stage in the hybrid embryos. Further functional analyses of the genes whose expression was misregulated in the hybrid blastoderms could facilitate the uncovering of the molecular basis of hybrid lethality.

## Supporting information

**S1 Data. Number of chromosomes in parental species and their hybrids.**
(XLSX)

**S2 Data. Count data obtained from mRNA sequencing.**
(TXT)

**S3 Data. List of 285 genes that were upregulated in parental species.**
(XLSX)

**S1 Fig. Stage X blastoderms of chicken, quail, and their hybrids used for mRNA sequencing.** Images of blastoderms are shown with their sample numbers.
(TIF)

**S2 Fig. Stage XIII/XIV blastoderms of chicken, quail, and their hybrids used for mRNA sequencing.** Images of blastoderms are shown with their sample numbers.
(TIF)

**S3 Fig. Gene expression changes from the stage X to the stage XIII/XIV in female embryos.** A–C. Comparison of gene expression changes [log2(fold change)] between quail ('Q') and

chickens ('G') (A), between quail ('Q') and quail-derived alleles in the hybrids ('HQ') (B), and between chickens ("G") and chicken-derived alleles in the hybrids ('HG') (C). Pearson's correlation efficient (r) is indicated above the graphs. D–F. Comparison of the direction of gene expression changes between quail and chickens (D), between quail and quail-derived alleles in the hybrids (E), and between chickens and chicken-derived alleles in the hybrids (F). The number in each rectangle indicates the percentage of genes. Percentages of genes that exhibited the same direction of expression changes are indicated in bold-lined rectangles. Numbers and percentages of genes exhibiting the same or different directions of expression change are shown in the tables. Color scale at the far right shows the percentage of genes.
(TIF)

**S4 Fig. Patterns of expression changes of 60 genes whose expression was misregulated in the male and/or female hybrids.** A, B. Patterns of expression changes of the 60 genes in male (A) and female (B) embryos.
(TIF)

**S5 Fig. Patterns of expression changes of primitive streak formation-related genes in female blastoderms.**
(TIF)

**S1 Table The number of cells with abnormal number of chromosomes.**
(PDF)

**S2 Table. Summary of patterns of gene expression changes.**
(PDF)

**S3 Table. Overrepresented GO-BP terms in 285 genes that were upregulated in male and/ or female embryos of parental species.**
(PDF)

**S4 Table. GO-BP terms including those related to primitive streak formation and chromosome segregation.**
(PDF)

**S5 Table. Result of GO term enrichment analysis of 285 genes that were upregulated in male and/or female embryos of parental species (primitive streak formation- and chromosome segregation-related GO-BP terms).**
(PDF)

**S6 Table. Summary of GO term enrichment analysis of 60 genes showing pattern D.**
(PDF)

## Acknowledgments

We thank Kiyomi Imamura, Terumi Horiuchi, Sumio Sugano, and Yutaka Suzuki (the University of Tokyo) for supporting RNA sequencing analysis. Chicken fertilized eggs were provided by the Avian Bioscience Research Center, Graduate School of Bioagricultural Sciences, Nagoya University through the National Bio-Resource Project (NBRP) 'Chicken/Quail' supported by the Ministry of Education, Culture, Sports, Science, and Technology (MEXT) and Japan Agency for Medical Research and Development (AMED), Japan.

## Author Contributions

**Conceptualization:** Satoshi Ishishita, Yoichi Matsuda.

**Formal analysis:** Satoshi Ishishita.

**Funding acquisition:** Yoichi Matsuda.

**Investigation:** Satoshi Ishishita, Keiji Kinoshita.

**Methodology:** Satoshi Ishishita, Shoji Tatsumoto, Keiji Kinoshita, Yasuhiro Go.

**Resources:** Keiji Kinoshita, Mitsuo Nunome, Takayuki Suzuki.

**Software:** Satoshi Ishishita, Shoji Tatsumoto, Yasuhiro Go.

**Supervision:** Yoichi Matsuda.

**Visualization:** Satoshi Ishishita.

**Writing – original draft:** Satoshi Ishishita.

**Writing – review & editing:** Satoshi Ishishita, Shoji Tatsumoto, Keiji Kinoshita, Mitsuo Nunome, Takayuki Suzuki, Yasuhiro Go, Yoichi Matsuda.

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
