## [Decision Letter · Decision Letter 0]

29 May 2020

PONE-D-20-09946

Transcriptome analysis revealed misregulated gene expression in blastoderms of interspecific chicken and Japanese quail F1 hybrids

PLOS ONE

Dear Dr. Matsuda,

Thank you for submitting your manuscript to PLOS ONE. After careful consideration, we feel that it has merit but does not fully meet PLOS ONE’s publication criteria as it currently stands. Therefore, we invite you to submit a revised version of the manuscript that addresses the points raised during the review process.

We look forward to receiving your revised manuscript.

Kind regards,

Jae Yong Han, Ph.D.

Academic Editor

PLOS ONE

Journal Requirements:

2. In your Methods section, please include a comment about the state of the animals following this research. Were they euthanized or housed for use in further research? If any animals were sacrificed by the authors, please include the method of euthanasia and describe any efforts that were undertaken to reduce animal suffering.

3. We note that you are reporting an analysis of a microarray, next-generation sequencing, or deep sequencing data set. PLOS requires that authors comply with field-specific standards for preparation, recording, and deposition of data in repositories appropriate to their field. Please upload these data to a stable, public repository (such as ArrayExpress, Gene Expression Omnibus (GEO), DNA Data Bank of Japan (DDBJ), NCBI GenBank, NCBI Sequence Read Archive, or EMBL Nucleotide Sequence Database (ENA)). In your revised cover letter, please provide the relevant accession numbers that may be used to access these data. For a full list of recommended repositories, see http://journals.plos.org/plosone/s/data-availability#loc-omics or http://journals.plos.org/plosone/s/data-availability#loc-sequencing.

Additional Editor Comments (if provided):

Reviewers' comments:

Reviewer's Responses to Questions

**Comments to the Author**

1. Is the manuscript technically sound, and do the data support the conclusions?

Reviewer #1: Partly

Reviewer #2: Yes

2. Has the statistical analysis been performed appropriately and rigorously? 

Reviewer #1: N/A

Reviewer #2: Yes

3. Have the authors made all data underlying the findings in their manuscript fully available?

Reviewer #1: Yes

Reviewer #2: Yes

4. Is the manuscript presented in an intelligible fashion and written in standard English?

Reviewer #1: Yes

Reviewer #2: Yes

5. Review Comments to the Author

Reviewer #1: Comments

In this manuscript, Ishishita et al describes that the numerical chromosomal abnormalities due to segregation failure does not cause the lethality of chicken–quail hybrid embryos, and that the developmental arrest at the preprimitive streak stage in the hybrids is mainly caused by the downregulated expression of the genes involved in various biological processes. The later part is the important part of this manuscript, however it raises several questions in its current form. Please see below for my specific comments on this manuscript.

Methods section:

In methods section, the chicken management, semen collection, and AI is not clear. Line 125 indicates that the chicken eggs and semen were supplied by the National BioResource Project…Line 131 indicates that the chickens were maintained (locally?). Line 135 indicates that the semen was collected just before AI…. If the chicken egg and semen were supplied, no need of chicken maintenance, as the paper just deals with chicken embryos. In this case, authors need to write semen preservation. If the semen was collected by authors, statement in line 125 should be excluded.

Lines 127-131. While both chromosome and gene expression analyses included in a single paper, it is not clear why Ehime-jidori chickens used for chromosome analysis only, and BL-E chickens used for gene expression analysis only.

Please expand the molecular sexing procedure. When and how the sexing of early embryos (stage X – XIII/XIV) detected?

Results section:

In figure 1, please use different arrows to indicate CJA-Bgl II-M9 signal in 1E, and quail-derived chromosomes in 1F.

In figure 4D, what does mean FC 1/2?

Lines 345-353. Please indicate clearly that you are describing male embryos here. I understood only after reading the corresponding figure legend (Fig. 4). The female counterpart (shown in S1 figure only) is not described and cited in this paragraph.

Lines 349-352. “The numbers of genes whose directions…” Please simply this sentence for clear understanding, and write the number of genes.

Line 371-376. “genes whose expression is upregulated …. referred to as pattern D)”. It is little difficult to connect this message, Fig. 5A, and S2 Fig. in the current form. I suggest to split the S2 Fig. with grouping of those 28 male genes, 28 female genes and 4 common genes fall on pattern D in Fig.5A.

Lines 370-409. This should be the major vital portion of the manuscript as the authors primarily claim in the title. However, it does not strengthen the manuscript, and raises many questions.

1. While avian/chicken specific GO terms are available in many databases including AmiGO, why the authors used Homo sapiens database as the reference? (stated in the methods section, lines 187-190).

2. What is the functional classification of genes listed in Fig. 5B, and how they differ from Fig.5D? does the other genes have no functional classification?

3. Except a very few term, most of the terms shown in Fig. 5B-D are not relevant to embryonic development at the analysis period, stage X – stage XIII/XIV. Authors need to screen terms closely related to stage X – XIII/XIV development. You may also include terms related to shortly before and after this time point. In addition, you can include the terms related to chromosomal properties to support the first section of this manuscript.

4. I suggest the authors to search and include the signaling pathways of those 60 misregulated genes and include as Fig. 5E. It will be interesting if you restrict the signaling pathways related to early embryonic development.

5. Then, Fig 5E can be moved to Fig 6. This information is also inadequate. Please extend this figure by showing a table containing critical/exact role of these markers during embryonic polarity before and during gastrulation.

Discussion section:

This section should be modified according to the revision of results section (lines 370-409, and Fig 5) after considering my review comments, primarily focusing on GO and signaling terms affected in stage X – XIII/XIV hybrid embryos.

Reviewer #2: The manuscript is about identifying the mis-regulated gene expression in blastoderms of interspecific chicken-Japanese quail F1 hybrids. Using transcriptome analysis, they have found that numerical abnormalities due to a segregation failure does not lead to the lethality of chicken–quail hybrid embryos, and that the developmental arrest at the pre-primitive streak stage in the hybrids is mainly caused by the downregulated expression of the genes involved in various biological processes such as translation and gastrulation.

The manuscript lacks the idiographic information in several parts and needs extensive restructuring.

Below are my specific comments mainly on the arrangement of materials and methods.

1. The manuscript is studying chicken (Male) and Japanese quail (Female) F1 hybrids. As well known, the fertilization and lethal rate of hybrids from the different combination is discrepancy. Have you looked at the gene expression of chicken (Female) and Japanese quail (Male) hybrids? If so, how about the result? If not, why?

2. Figures included in this manuscript are too blurry to look at the details. Please reupload figures with high resolution, especially Figure 1 and Figure 5.

3.The Figure legends in this manuscript should be corrected.

4. The statistics analysis needs to be corrected. Why were two different statistic methods used in the same analysis? For example, in Figure 2: the Tukey-Kramer test was used in panel A and B, while the Dunnet’s test was used in panel C and D. Besides, please explain why not use the Pearson’s test but Spearman’s rank in correlation coefficient analysis in Figure 4.

5. In Figure 2A, how do you explain the outliers in the Hybird group in panel A? In addition, the “A” is smaller than other alphabets, please correct the format of figure.

6. Discussion needs to be restructured explaining more about the impact of current findings in order to make the paper compelling for readers.

6. PLOS authors have the option to publish the peer review history of their article (what does this mean?). If published, this will include your full peer review and any attached files.

Reviewer #1: Yes: Deivendran Rengaraj

Reviewer #2: No

---

## [Author Response · Author response to Decision Letter 0]

13 Sep 2020

Response to Reviewer #1

We wish to express our appreciation to the Reviewer for his or her insightful comments, which have helped us significantly improve the paper.

Methods section:

In methods section, the chicken management, semen collection, and AI is not clear. Line 125 indicates that the chicken eggs and semen were supplied by the National BioResource Project…Line 131 indicates that the chickens were maintained (locally?). Line 135 indicates that the semen was collected just before AI…. If the chicken egg and semen were supplied, no need of chicken maintenance, as the paper just deals with chicken embryos. In this case, authors need to write semen preservation. If the semen was collected by authors, statement in line 125 should be excluded.

Response: Line 117–118. We have removed this part because we used semen for artificial insemination immediately after collection of semen from the strain Ehime-jidori. We have also corrected Acknowledgement as follows. “Chicken semen and fertilized eggs were provided by the Avian Bioscience Research Center,” has been changed to “Chicken fertilized eggs were provided by the Avian Bioscience Research Center,” (Line 525¬–526).

Lines 128-131. While both chromosome and gene expression analyses included in a single paper, it is not clear why Ehime-jidori chickens used for chromosome analysis only, and BL-E chickens used for gene expression analysis only.

Response: Line 123–125. The two analyses were conducted at different times using different lines because of the availability of adult chickens when the experiments were carried out.

Please expand the molecular sexing procedure. When and how the sexing of early embryos (stage X ? XIII/XIV) detected?

Response: Line 148–158. Embryonic tissues, which were collected at stage X to XIII/XIV, were minced, and a part of each tissue was used for molecular sexing. The remaining cell suspensions were used for RNA extraction. The description for the molecular sexing method has been included in the item of mRNA sequencing in the revised manuscript. 

Results section:

In figure 1, please use different arrows to indicate CJA-Bgl II-M9 signal in 1E, and quail-derived chromosomes in 1F.

Response: Line 258–267. We have corrected Fig 1D–F and the corresponding figure caption for readability. Microchromosomes that were hybridized only with CJA-BglII-M9 were considered quail-derived chromosomes and are indicated by asterisks in Fig 1F.

In figure 4D, what does mean FC 1/2?

Response: Line 182–188. Fold change (FC) of gene expression of stage XIV embryos was calculated by comparing with that of stage X embryos. When the expression level of stage XIV embryos is less than half that of stage X embryos, it has been described as FC is lower than 1/2 (FC < 1/2). We have added description regarding this issue in Methods. “1/2” has been change to “0.5”.

Lines 345-353. Please indicate clearly that you are describing male embryos here. I understood only after reading the corresponding figure legend (Fig. 4). The female counterpart (shown in S1 figure only) is not described and cited in this paragraph.

Response: Line 367–369. We have added the underlined phrase and sentence in this part as follows:

The correlation coefficients of the expression change were much higher in G vs. HG (0.502) and Q vs. HQ (0.558) than in G vs. Q (0.124) in males (Fig 4A–C). Similar results were also obtained from transcriptome analysis of female embryos (S1 Fig).

Lines 349-352. “The numbers of genes whose directions…” Please simply this sentence for clear understanding, and write the number of genes.

Response: Line 3723–377. We have corrected this part as follows:

We revealed that 8,376 (72.4%) out of a total of 11,575 genes exhibited similar changes in expression between quail (Q) and chickens (G) (Fig 4E). We also showed that 9,253 (79.9%) and 8,572 (74.1%) genes exhibited similar changes in expression between quail (Q) and hybrids (quail-derived alleles, HQ) and between chickens (G) and hybrids (chicken-derived alleles, HG), respectively, in males (Fig 4F and G). The numbers were higher than that between parental species. Similar results were also obtained in females (S1 Fig).

Line 371-376. “genes whose expression is upregulated …. referred to as pattern D)”. It is little difficult to connect this message, Fig. 5A, and S2 Fig. in the current form. I suggest to split the S2 Fig. with grouping of those 28 male genes, 28 female genes and 4 common genes fall on pattern D in Fig.5A.

Response: As suggested by the reviewer, we have split the data in S2 Fig and classified 28 male genes, 28 female genes and 4 common genes into pattern D.

Lines 370-409. This should be the major vital portion of the manuscript as the authors primarily claim in the title. However, it does not strengthen the manuscript, and raises many questions.

1. While avian/chicken specific GO terms are available in many databases including AmiGO, why the authors used Homo sapiens database as the reference? 

Response: Line 189–195. Fig 6B, C and S3, 5, and 6 Tables.

We used the Gene Ontology (GO) Annotation database for human for the reason that the GO database is more substantial in human compared with that of the chicken. We have carried out GO term enrichment analysis again using the chicken database according to the reviewer’s comment. We have used PANTHER classification system, but not DAVID in this analysis because PANTHER is used in AmiGO. The result of the overrepresentation test for GO terms of biological process is shown in Fig 6B, C and S3, 5, and 6 Tables.

2. What is the functional classification of genes listed in Fig. 5B, and how they differ from Fig.5D? does the other genes have no functional classification?

Response: We have deleted the gene list shown in Fig. 5B of the original manuscript because the functional classification analysis is specific to DAVID.

3. Except a very few term, most of the terms shown in Fig. 5B-D are not relevant to embryonic development at the analysis period, stage X ? stage XIII/XIV. Authors need to screen terms closely related to stage X ? XIII/XIV development. You may also include terms related to shortly before and after this time point. In addition, you can include the terms related to chromosomal properties to support the first section of this manuscript.

Response: Line 400–403, 417–435. Fig 6C and S3–6 Tables.

No primitive streak formation-related GO terms came up by the overrepresentation test in the present study. However, as suggested by the reviewer, we picked up GO biological process terms related to primitive streak formation and chromosome segregation and show them and relevant genes in Fig. 6C and S4–6 Tables. Overrepresented GO terms of molecular function categories (shown in Fig. 5D of the original manuscript) have been removed because the result is redundant with that of the overrepresentation test regarding GO terms of biological process categories. We also performed GO analysis of 285 genes that were upregulated in male and/or female embryos of parental species and show the result in S3 Table.

Response: We could not find the terms related to shortly before and after this time point. 

Response: Line 425¬–426. We have added the following sentence.

Consistent with the results of chromosome analysis in the present study, none of the 60 genes were associated with mitotic chromosome segregation-related GO-BP categories.

4. I suggest the authors to search and include the signaling pathways of those 60 misregulated genes and include as Fig. 5E. It will be interesting if you restrict the signaling pathways related to early embryonic development.

Response: We have conducted the overrepresentation test regarding signaling pathways. We noticed that there were no categories associated with early embryonic development, such as mesoderm formation and gastrulation, in the reference database.

5. Then, Fig 5E can be moved to Fig 6. This information is also inadequate. Please extend this figure by showing a table containing critical/exact role of these markers during embryonic polarity before and during gastrulation.

Response: Line 436–440. This data has been shown in Fig 6 following the reviewer’s suggestion. We have picked up nine marker genes that are well-known for being functional before, during and after the formation of primitive streak. We have added a table describing functions of these markers in Fig 6. Furthermore, we made discussion about this issue in line 486-493.

Discussion section:

This section should be modified according to the revision of results section (lines 370-409, and Fig 5) after considering my review comments, primarily focusing on GO and signaling terms affected in stage X ? XIII/XIV hybrid embryos.

Response: We have revised the Discussion section. First, we have added brief description for the aim and significance of this study as the first paragraph (line 451–456). According to the reviewer’s suggestion, we have discussed biological processes that may affect the development of hybrid embryos, focusing on overrepresented GO terms of biological process categories (line 466–478) and GO terms of biological process categories that are related to primitive streak formation (line 479–485). 

We wish to thank the Reviewer again for his or her valuable comments.

 

Response to Reviewer #2

We wish to express our appreciation to the Reviewer for his or her insightful comments, which have helped us significantly improve the paper.

1. The manuscript is studying chicken (Male) and Japanese quail (Female) F1 hybrids. As well known, the fertilization and lethal rate of hybrids from the different combination is discrepancy. Have you looked at the gene expression of chicken (Female) and Japanese quail (Male) hybrids? If so, how about the result? If not, why?

Response: Because of a low amount of semen that can be obtained from male quail, generation of F1 hybrids between chicken females and quail males by artificial insemination is difficult. To our knowledge, no one has succeeded. 

2. Figures included in this manuscript are too blurry to look at the details. Please reupload figures with high resolution, especially Figure 1 and Figure 5.

Response: As suggested, we have reuploaded high-resolution figures. In addition, Fig 1 has been divided into two Figures (Fig 1 and Fig 2 in the revised manuscript)

3.The Figure legends in this manuscript should be corrected.

Response: We have corrected the figure legends.

4. The statistics analysis needs to be corrected. Why were two different statistic methods used in the same analysis? For example, in Figure 2: the Tukey-Kramer test was used in panel A and B, while the Dunnet’s test was used in panel C and D. Besides, please explain why not use the Pearson’s test but Spearman’s rank in correlation coefficient analysis in Figure 4.

Response: Line 230–232, line 304–311, line 319–327, and line 367¬–368. As the reviewer pointed out, Tukey-Kramer test was conducted for both panel A and B, and no significant difference was detected. We have also conducted Pearson’s tests and Fig 5 and S3 Fig shows the results of Pearson’s tests.

5. In Figure 2A, how do you explain the outliers in the Hybird group in panel A? In addition, the “A” is smaller than other alphabets, please correct the format of figure.

Response: The outliers may be caused by the artifact when chromosome spreads were prepared, which cause the loss of chromosomes. We have corrected the size of the letter “A” in Fig 3 (Fig 2 in the original manuscript). 

6. Discussion needs to be restructured explaining more about the impact of current findings in order to make the paper compelling for readers.

Response: We have revised the Discussion section as follows. 

We have added brief description for the aim and significance of this study and as the first paragraph (line 451–456). Subsequent to the discussion about chromosome analysis (line 456–465), we have discussed about biological processes that may affect the development of hybrid embryos, based on overrepresented GO biological process categories (line 466–479), expression of genes associated with GO biological process categories that are related to primitive streak formation (line 479–485), and expression of primitive streak formation-related genes (line 486-493). We have made discussion about four genes that are misregulated in hybrids of both sexes (494–501). Then, we have made the discussion regarding the cause of misregulatd gene expression and future perspective for elucidating the molecular basis of embryonic lethality in chicken-quail hybrids (502–511). 

We wish to thank the Reviewer again for his or her valuable comments.

---

## [Decision Letter · Decision Letter 1]

22 Sep 2020

Transcriptome analysis revealed misregulated gene expression in blastoderms of interspecific chicken and Japanese quail F1 hybrids

PONE-D-20-09946R1

Dear Dr. Matsuda,

We’re pleased to inform you that your manuscript has been judged scientifically suitable for publication and will be formally accepted for publication once it meets all outstanding technical requirements.

Kind regards,

Jae Yong Han, Ph.D.

Academic Editor

PLOS ONE

Additional Editor Comments (optional):

Reviewers' comments:

Reviewer's Responses to Questions

**Comments to the Author**

1. If the authors have adequately addressed your comments raised in a previous round of review and you feel that this manuscript is now acceptable for publication, you may indicate that here to bypass the “Comments to the Author” section, enter your conflict of interest statement in the “Confidential to Editor” section, and submit your "Accept" recommendation.

Reviewer #1: All comments have been addressed

Reviewer #2: All comments have been addressed

2. Is the manuscript technically sound, and do the data support the conclusions?

Reviewer #1: Yes

Reviewer #2: No

3. Has the statistical analysis been performed appropriately and rigorously? 

Reviewer #1: Yes

Reviewer #2: No

4. Have the authors made all data underlying the findings in their manuscript fully available?

Reviewer #1: Yes

Reviewer #2: Yes

5. Is the manuscript presented in an intelligible fashion and written in standard English?

Reviewer #1: Yes

Reviewer #2: Yes

6. Review Comments to the Author

Reviewer #1: (No Response)

Reviewer #2: Although the author's answer to the first question is vague, it does not affect the quality of the whole manuscript . In addition, I focused on the quality of the Figures in the manuscript, which were too blurry. Hope the author can provide clear original figures.

7. PLOS authors have the option to publish the peer review history of their article (what does this mean?). If published, this will include your full peer review and any attached files.

Reviewer #1: **Yes: **Deivendran Rengaraj

Reviewer #2: No

---

## [Editor Report · Acceptance letter]

1 Oct 2020

PONE-D-20-09946R1 

Transcriptome analysis revealed misregulated gene expression in blastoderms of interspecific chicken and Japanese quail F_1_ hybrids 

Dear Dr. Matsuda:

I'm pleased to inform you that your manuscript has been deemed suitable for publication in PLOS ONE. Congratulations! Your manuscript is now with our production department. 

Kind regards, 

on behalf of

Prof. Jae Yong Han 

Academic Editor

PLOS ONE